# DR-MMSearchAgent: Deepening Reasoning in Multimodal Search Agents

**Shengqin Wang** [1]   **Wentao Yan** [1]   **Huichi Zhou** [2 3]   **Yihang Chen** [2 3]   **Kun Shao** [4]   **Zhizhong Zhang** [1]   **Yuan Xie** [1 5]

## Abstract

Agentic multimodal models have garnered significant attention for their ability to leverage external tools to tackle complex tasks. However, it is observed that such agents often meet premature interaction collapse, caused by two primary reasons: 1) the terminal reward often appending on the last token prevents the advantage from distinguishing trajectories with exploratory behavior; 2) excessively redundant context hinders the agent from absorbing useful feedback. To address these issues, we propose the Deepening Reasoning MMSearchAgent, the framework leverages the structural proximity to derive advantage signals from the whole rollout trajectories in an entire batch, such that trajectories of different lengths are further encouraged to be generated, even when containing the same correct answer. Additionally, differentiated gaussian rewards are employed to dynamically calibrate interaction tolerance, thereby ensuring information reliability and reducing redundancy. To support multi-turn interaction training, we have constructed a multi-step deep-reasoning dataset including 3602 high-quality QA pairs with at least 3 reasoning steps. Extensive experiments demonstrate that our method achieves state-of-the-art performance, outperforming the MMSearch-R1 by 8.4% on FVQA-test.

## 1. Introduction

Agentic multimodal models, when addressing time-sensitive Visual Question Answering (VQA), demonstrate the capability of leveraging the search tools to acquire comprehensive knowledge. Previous approaches have largely relied on retrieval-augmented generation (RAG) methods (Chen

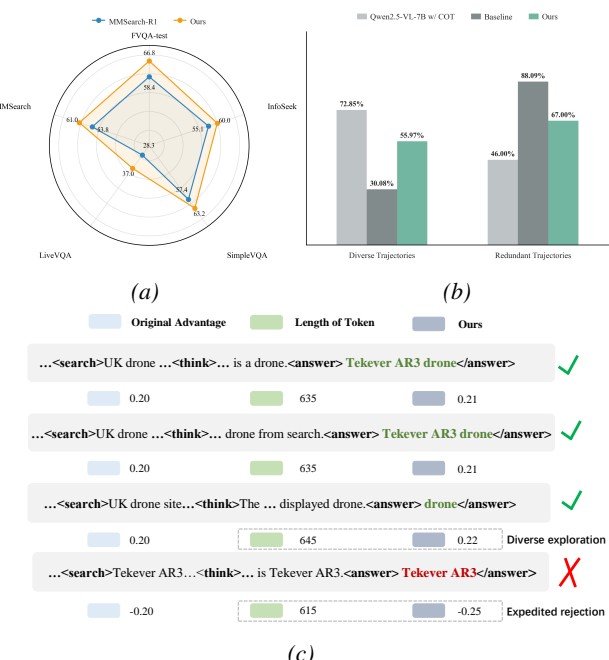

*Figure 1.* (a) Radar chart of comparisons on accuracy. (b) Comparison of trajectory diversity within the same reward and redundancy (non-zero reward). Diverse Trajectories denote the percentage of trajectories with varying response lengths under identical rewards, while Redundant Trajectories represent the percentage of trajectories with duplicated rewards and identical response lengths. (c) Comparison of our advantage calculation method with current methods.

et al., 2022; 2024b; Yu et al., 2024) and early-stage search agents (Hu et al., 2023; Jiang et al., 2024; Zhang et al., 2024b). While these methods excel at integrating static external knowledge, they struggle to capture complete time-sensitive information in dynamic multimodal environments, leading to ineffective searches. To bridge this gap, recent studies (Wu et al., 2025; Narayan et al., 2025) have successfully integrated external search tools and employed reinforcement learning to train agents capable of multi-turn reasoning.

However, existing RL-based multimodal search methods suffer from exploration degradation. For instance, MMSearch-R1 (Wu et al., 2025) tend to converge toward suboptimal two-turn tool interactions, which limits their capacity to tackle complex, multi-step tasks. This failure is primar-

---

[1]East China Normal University [2]University College London [3]Huawei Noah's Ark Lab [4]Independent Researcher [5]Shanghai Innovation Institute. Correspondence to: Yuan Xie <yxie@cs.ecnu.edu.cn>.

*Proceedings of the 43rd International Conference on Machine Learning*, Seoul, South Korea. PMLR 306, 2026. Copyright 2026 by the author(s).

ily twofold. First, existing advantage estimation overlook the verification of reward positional information from a global perspective by comparing the whole roll-out trajectories of different lengths, resulting in coarse-grained signals that lack the diversity. Specifically, as shown in Figure 1 (b), in the early stages of model training, the trajectories of Chain-of-Thought (CoT) reasoning exhibit high diversity. However, as training progresses, the proportion of diverse trajectories shrinks over time while repetitive ones proliferate, causing internal advantage signals to lose their discriminative power. Second, trajectory redundancy and contextual interference prevent further for exploration. As depicted in the left panel of Figure 4, models frequently handle long-sequence responses from external tools. Such informational erosion causes the agent to prematurely terminate reasoning or bypass essential multi-turn tool calls when facing complex objectives.

To address these issues, as in Figure 2, we propose the Deepening Reasoning MMSearchAgent (DR-MMSearchAgent). First, we introduce the Structural Proximity-weighted Advantage Injection (SPAI), that quantifies the structural proximity between verification reward trajectories. It therefore effectively distinguishes heterogeneous trajectories under the same reward, but with different CoT length, enhancing the diversity of advantage signals. As shown in Figure 1 (b), (c), by incorporating the positional information of verifiable rewards, this mechanism mitigates the decline in reward trajectory diversity and the redundancy of repetitive trajectories, thereby ensuring exploration of the model.

Second, we design a Bidirectionally Guided Adaptive Smoothed (BGAS) reward. Utilizing differentiated gaussian reward distributions, this component guides the model to dynamically calibrate its exploration based on the quality of intermediate solutions, thereby ensuring the credibility of interaction behaviors. Furthermore, a refining agent is integrated to perform real-time compression and denoising of redundant interaction trajectories. Our contributions are summarized below:

**Co-optimized Deep Reasoning Method.** The method considers the positional information of verification rewards via a structural proximity weighting mechanism. It encourages to generate trajectories with different CoT length to promote exploration. Furthermore, it integrates a bidirectional-guided adaptive smoothing reward mechanism to dynamically calibrate the model's interaction flow, achieving reliable deep reasoning.

**Multi-hop Reasoning MM-search Framework.** We have constructed a multi-step deep-reasoning dataset from the latest internet visual information. Simultaneously, a refinement agent is integrated to perform real-time compression and denoising of redundant interaction trajectories.

**Performance Improvement.** Our framework sets a new state-of-the-art record on existing knowledge-intensive benchmarks while demonstrating leading capabilities on other fundamental vision tasks.

## 2. Related Work

**Multimodal Large Language Models.** Early MLLM research achieved cross-modal alignment primarily through lightweight adapters or projection layers linking pre-trained visual encoders to LLMs (Liu et al., 2024; Chen et al., 2024a; Liu et al., 2023). Models like Qwen2.5-VL (Bai et al., 2025), InternVL3 (Zhu et al., 2025a), and Qwen3 (Yang et al., 2025a) significantly improved accuracy on tasks such as VQA and image-text description generation by expanding training datasets and introducing diverse visual inputs (Wu et al., 2025; Wang et al., 2025b; Zhang et al., 2024a).

**Reinforcement Learning for Large Language Models.** Reinforcement learning has become a central technique for aligning LLMs toward desired reasoning behaviors. Early methods such as PPO-based RLHF (Schulman et al., 2017; Shao et al., 2024) focus primarily on preference optimization, while later frameworks (Jaech et al., 2024; Guo et al., 2025; Lin et al., 2024) aim to improve stability, sample efficiency, or reward smoothness. Recent studies (Chen et al., 2022; 2024b; Yu et al., 2024) on tool-augmented LLMs explore how RL can encourage tool selection and planning.

## 3. Method

### 3.1. Overview

Figure 2 illustrates the architecture of DR-MMSearchAgent. The Advantage Injection module presents a specific implementation of SPAI, which models the structural variance of reward tokens based on response length and reward magnitude, thereby further enhancing the diversity of advantage signals. Meanwhile, the Adaptive Reward module incorporates a Bidirectionally Guided Adaptive Smoothing approach. By leveraging smooth gaussian curves, it dynamically calibrates the efficiency and search depth of the model's intermediate reasoning steps. Furthermore, we constructed a BridgeVQA dataset anchored on visual elements to facilitate training. The extracted textual and visual information is integrated into a Tool Service, while a Refining Agent is employed to compress redundant tool contexts.

### 3.2. Structural Proximity-weighted Advantage Injection

**GRPO in MLLMs.** Group-Relative Policy Optimization (GRPO) (Shao et al., 2024) stabilizes training by introducing an intra-group comparison mechanism. For a given query $q$, the policy generates a group of $N$ responses $O = \{o_1, \ldots, o_N\}$, and then the reward $R$ across this entire

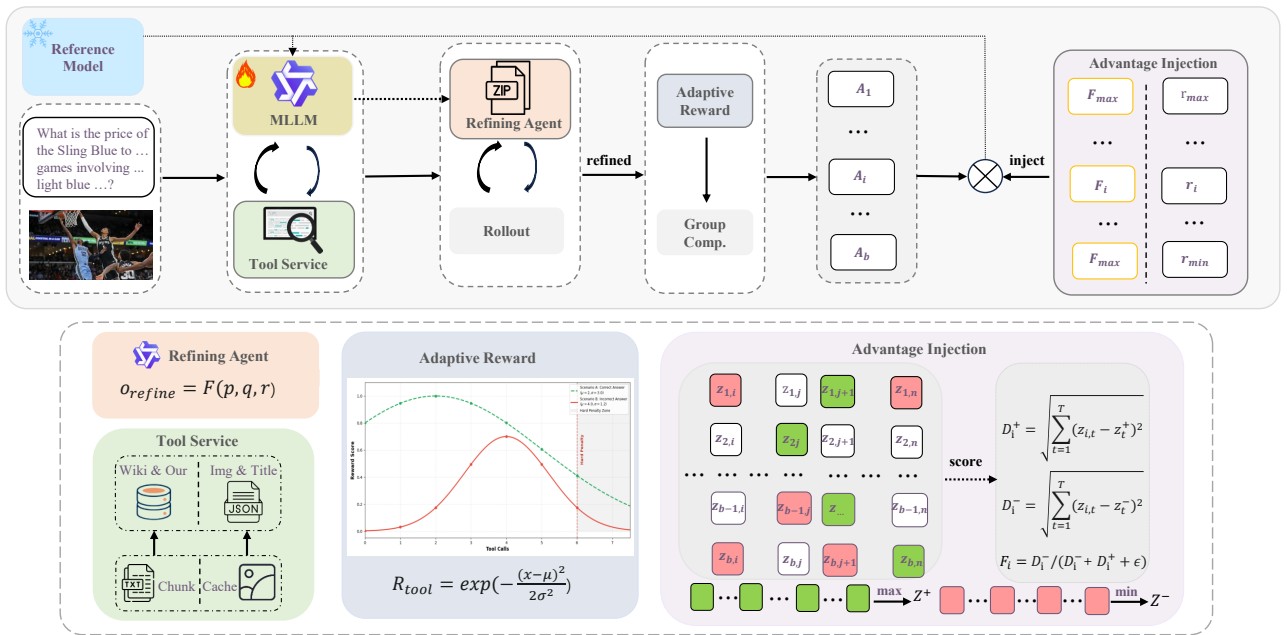

*Figure 2.* Framework for DR-MMSearchAgent. The upper panel illustrates the overall training framework, while the lower panel depicts the advantage injection mechanism, the adaptive reward mechanism, the refining agent, and the updatable multimodal tool services.

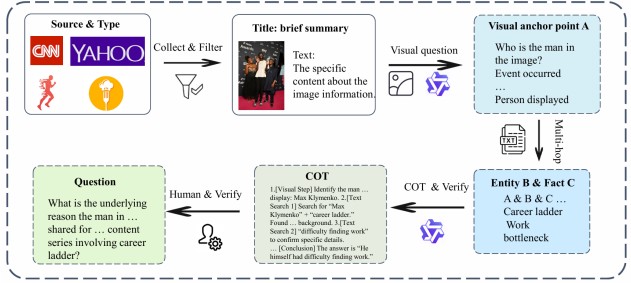

*Figure 3.* Schematic of the BridgeVQA construction workflow, featuring data collection, multi-hop problem generation, and dual-phase filtering.

group is computed. Its core objective formula is:

$$E_{i,t}\left[\min\left(\rho_t^{(i)}A^{(i)}, \text{clip}A^{(i)}\right)\right] - B, \quad (1)$$

where $B = \beta KL(\pi_\theta\|\pi_{\text{ref}})$, $A^{(i)} = \frac{r(\tau)-\text{mean}(R)}{\text{std}(R)}$.

In the advantage, $r(\tau)$ denotes the reward appending to the last response token, and $R$ represents the reward matrix within the whole roll-out trajectories. In its implementation, the advantage is only calculated in the terminal scalar of the verification reward, failing to account for the specific positional information of trajectory rewards.

Based on the verification reward, we construct $R' \in \mathbb{R}^{G \times T}$, where $G$ denotes the group size and $T$ represents the maximum sequence length. For each trajectory, a reward value is assigned only to the last token of the response $L_i$, with all other positions set to zero:

$$r_{i,t} = \begin{cases} r(\tau), & \text{if } t = L_i \\ 0, & \text{if } 1 \le t < L_i. \end{cases} \quad (2)$$

The $R'$ provide the positional information or CoT length of reward $r(\tau)$ by considering the whole roll-out trajectories in an entire batch. Notably, as illustrated in Figure 1 (b), through $R'$, a pronounced isomorphism phenomenon appears where reward tokens at disparate positions are frequently assigned identical terminal scalar rewards. This redundancy in advantage signals obviously obscures the structural disparities between trajectories, and therefore diminishes the diversity of advantage signals, ultimately resulting in insufficient exploration.

**Specific Implementation of SPAI.** We draw inspiration from the Technique for Order Preference by Similarity to Ideal Solution (TOPSIS) (Hwang & Yoon, 1981)—a method whose core lies in measuring a sample's proximity to an ideal state by synthesizing various features. Integrating this philosophy, we propose the Structural Proximity-weighted Advantage Injection (SPAI) method. Specifically, SPAI leverages response length and reward magnitude as key indicators to capture the structural differences of reward tokens within the Chain-of-Thought (CoT) from a global perspective. By doing so, this approach explicitly differentiates distinct reasoning trajectories, thereby effectively increasing the diversity of advantage signals and promoting agent exploration. We provide further technical details regarding

this method in Appendix A.

In our implementation, given $R'$, we define the effective group size $G = \text{batch} \times \text{rollout}$ to ensure statistically robust trajectory modeling. As shown in the Advantage Injection of Figure 2. First, to handle numerical instability in sparse reward, we apply a controlled normalization to obtain $z_{i,t}$:

$$z_{i,t} = \frac{r_{i,t}}{\|\mathbf{r}_t\|_2 + \mathbb{1}_{\{\|\mathbf{r}_t\|_2 = 0\}}}, \quad (3)$$

where $\|\mathbf{r}_t\|_2 = \sqrt{\sum_{j=1}^{G} r_{j,t}^2}$ and:

$$\mathbb{1}_{\{\|\mathbf{r}_t\|_2 = 0\}} = \begin{cases} 1, & \text{if } \|\mathbf{r}_t\|_2 = 0 \\ 0, & \text{if } \|\mathbf{r}_t\|_2 > 0 \end{cases}. \quad (4)$$

Next, as shown at the bottom of the Advantage Injection module in Figure 2, we construct a virtual positive ideal solution $Z^+$ and a negative ideal solution $Z^-$—which physically represent the upper and lower bounds of reward magnitudes across varying response lengths, reflecting the diversity profile of the entire batch—by aggregating the extrema at each timestep $t$ in the trajectory sequence:

$$Z^+ = (z_1^+, \ldots, z_T^+), \quad Z^- = (z_1^-, \ldots, z_T^-), \quad (5)$$

where $z_t^+ = \max_i\{z_{i,t}\}$ and $z_t^- = \min_i\{z_{i,t}\}$. Note that these ideal solutions are virtual reward boundaries, encoding the global CoT length information rather than a single trajectory. We then compute the euclidean distances $D_i^+, D_i^-$ between the current trajectory $i$ and these largest and smallest solutions:

$$D_i^+ = \sqrt{\sum_{t=1}^{T}(z_{i,t} - z_t^+)^2}, \quad D_i^- = \sqrt{\sum_{t=1}^{T}(z_{i,t} - z_t^-)^2}. \quad (6)$$

The structure relative closeness score $F_i \in (0, 1)$ is derived as:

$$F_i = \frac{D_i^-}{D_i^+ + D_i^- + \epsilon}, \quad (7)$$

where a higher $F_i$ indicates a trajectory with a superior diversity score.

To foster diversified exploration, we employ an asymmetric injection mechanism. Let $F_{max} = \max_i F_i$ be the maximum structural score in the current batch. Finally, as in Figure 2, we modulate the original advantage $A(\tau)$ to produce the injected advantage:

$$A'(\tau) = A(\tau)(1 + W_i), \quad (8)$$

where the weighting factor $W_i$ is defined as:

$$W_i = \begin{cases} F_{max}, & \text{if } i \in \mathcal{I}_{min} \\ F_i, & \text{if } i \notin \mathcal{I}_{min} \end{cases}. \quad (9)$$

Here, $\mathcal{I}_{min}$ denotes the bottom $N\%$ of samples ranked by reward $r(\tau)$. Since low-quality trajectories typically yield negative advantages ($A(\tau) < 0$), we couple their weights with the maximum spatial score $F_{max}$ to further amplify these negative signals into a modified advantage $A'(\tau)$. This mechanism compels the policy to rapidly reallocating probability mass to accelerate exploration in higher-reward areas. To ensure stability, $N$ is chosen conservatively to target only tail-end samples that significantly deviate from expectations, thereby facilitating faster convergence while preventing oscillations in advantage estimation.

### 3.3. Bidirectionally Guided Adaptive Smoothing

Current reward functions in agentic frameworks predominantly focus on outcome-based feedback and formatting compliance, yet they lack explicit mechanisms to incentivize or regulate deep exploration. To address this, we decompose our reward signal into three distinct components: accuracy ($R_{\text{accuracy}}$), formatting compliance ($R_{\text{format}}$), and a bidirectionally guided tool efficiency score ($R_{\text{tool\_efficiency}}$).

$R_{\text{accuracy}} \in \{0, 1\}$ denotes a binary reward indicating the correctness of the inferred answer. $R_{\text{format}}$ is divided into two parts, primarily focusing on tool formatting and summarization standards to regulate the model's behavior during inference. $R_{\text{tool\_efficiency}}$ is designed to dynamically regulate interaction depth. We propose two distinct strategic regimes to balance brevity and exhaustive search:

**Efficiency-First Regime (When $R_{\text{accuracy}}$ is Correct).** When the model's reasoning aligns with the ground truth, the reward should penalize unnecessary tool calls to prevent reasoning bloat. In this regime, we aim for a distribution with a lower mean ($\mu_1$) and broader variance ($\sigma_1$), ensuring the model maintains conciseness when it is already on the right track.

**Exploration-Driven Regime (When $R_{\text{accuracy}}$ is Incorrect).** Conversely, if the initial reasoning fails, we hypothesize that the failure stems from insufficient evidence. Here, the reward mechanism shifts to encourage deeper exploration by targeting a higher number of tool calls ($\mu_2 > \mu_1$) with a narrower focus ($\sigma_2 < \sigma_1$). This subtly guides the agent to overcome exploration myopia and retrieve diverse information in subsequent iterations. Given a scaling factor that preventing reward hacking.

To mathematically formalize this adaptive behavior and avoid optimization instability in sparse-reward settings, we leverage a gaussian kernel function as a robust structural prior for $R_{\text{tool\_efficiency}}$:

$$G(N_{\text{tool}}; \mu, \sigma) = \exp\left(-\frac{(N_{\text{tool}} - \mu)^2}{2\sigma^2}\right). \quad (10)$$

The parameters are dynamically switched between $\Theta_1 =$

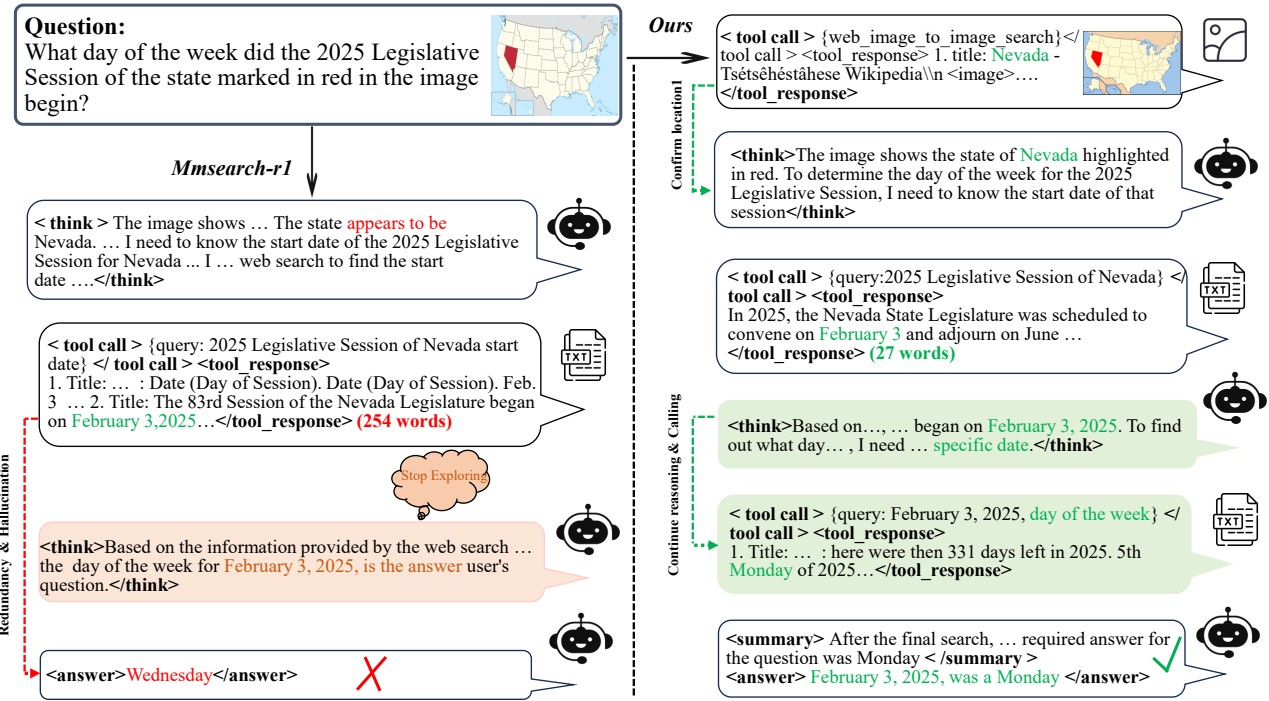

*Figure 4.* The comparison of multi-step interaction. Unlike traditional baseline models that exhibit short-sighted exploration and redundant trajectories, our framework enables stable and reliable interactions. It achieves further performance improvements while increasing the number of model interactions and reducing their corresponding lengths.

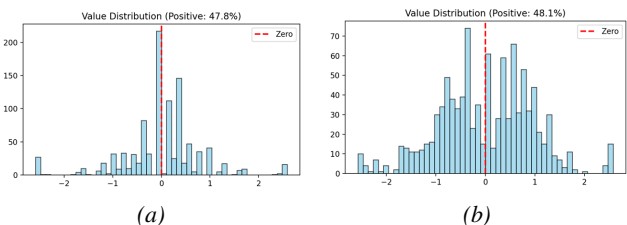

*Figure 5.* Advantage distribution in a training batch of baseline and our framework.

$(\mu_1, \sigma_1)$ and $\Theta_2 = (\mu_2, \sigma_2)$ based on the correctness of $R_{\text{accuracy}}$. These distributions serve not as hyperparameters for fine-tuning, but as fixed inductive biases (Table 5) that enable the agent to calibrate interaction depth via feedback.

Finally, these components are aggregated into the global reward function to ensure that accuracy remains the primary optimization objective:

$$R_{\text{final}} = 0.7 \cdot R_{\text{accuracy}} + 0.2 \cdot R_{\text{format}} + 0.1 \cdot R_{\text{tool\_efficiency}}. \quad (11)$$

### 3.4. The Framework of Deepening Reasoning MMSearch

Current knowledge-intensive training sets often suffer from limited search depth; for instance, the FVQA dataset in MMSearch-R1 was primarily designed for a maximum of two-turn tool interactions. Furthermore, these datasets frequently lack temporal relevance. To bridge this gap, we introduce BridgeVQA, a deep multi-hop VQA dataset sourced from the latest web information. Comprising 3,602 questions (detailed in Appendix D and E), BridgeVQA presents a significantly higher difficulty level than FVQA. Developed through a visual anchor pipeline, it integrates cross-modal dependencies and CoT mechanisms to address high-order reasoning challenges.

**The Pipeline of BridgeVQA.** As shown in Figure 3, we first systematically leverage the latest news website resources (Fu et al., 2025) to obtain image-text pairs. Then we perform visual entity anchoring. We locally deploy Qwen2.5-VL-72B-Instruct (Bai et al., 2025) to analyze acquired images and generate visual entity anchors. These anchors primarily include key events, persons, or specific objects within the images, serving as visual cues for subsequent question construction.

To guarantee an implicit reasoning path of at least three hops, we employ a structured question generation formula guided by strict logical constraints. First, the core query structure follows the triple dependency chain: Query (Anchor A) $\xrightarrow{\text{Relation } \mathcal{R}}$ Bridge B $\xrightarrow{\text{Dependency}}$ Target Fact C. This translates into the template: What is the [Target Fact C] of the [Bridge Entity B] that has a [Relation $\mathcal{R}$] with the [Visual

*Table 1.* Main results across five benchmarks. [§] Since this study utilizes the latest official version of LiveVQA, results from earlier versions used by WebWatcher are not strictly comparable.

| Model | Average | FVQA-test | InfoSeek | SimpleVQA | LiveVQA | MMSearch |
|---|---|---|---|---|---|---|
| *Direct* | | | | | | |
| GPT-4o | 36.0 | 41.7 | 42.7 | 46.6 | 26.9 | 22.2 |
| Gemini2.5 Pro | 36.4 | 37.2 | 37.0 | 53.4 | 27.7 | 26.9 |
| Qwen2.5-VL-7B | 18.1 | 20.9 | 23.9 | 30.4 | 8.3 | 7.2 |
| Qwen2.5-VL-32B | 25.0 | 24.7 | 25.8 | 40.1 | 18.7 | 15.7 |
| *Search Agent* | | | | | | |
| Qwen2.5-VL-7B | 26.0 | 34.2 | 28.3 | 35.8 | 10.7 | 21.1 |
| Qwen2.5-VL-32B | 38.0 | 51.3 | 38.0 | 48.5 | 24.8 | 27.3 |
| MMSearch-R1 | 50.6 | 58.4 | 55.1 | 57.4 | 28.3 | 53.8 |
| DeepMMSearch-R1 | — | — | 47.5 | 55.9 | — | — |
| WebWatcher[§] | — | — | — | 54.3 | — | 49.1 |
| DeepEyesV2 | — | 60.6 | 51.1 | 59.4 | — | **63.7** |
| Ours | **57.6** | **66.8** | **60.0** | **63.2** | **37.0** | 61.0 |

Anchor A] ? Then for complex multi-hop VQA instance generation. LLMs are utilized to generate preliminary question-answer pairs based on the formula. Crucially, each instance is paired with a reliable, interactive CoT trace, explicitly documenting the A → B → C reasoning path.

We established a rigorous data curation pipeline to extract high-quality instances. Initially, a preliminary screening was conducted using the model, where generated CoT sequences served as a basis for logical validation. Subsequently, for ambiguous cases, manual filtering and refinement were performed to resolve uncertainties. This pipeline ultimately yielded approximately 3,602 complex multi-hop instances, ensuring high logical consistency and verifiability.

**Compressed Refining Agent.** In Figure 4, we observe that text searches often yield prolix responses, leading to extensive interaction trajectories. To address this, we restrict retrieval to the top-3 results (see Figure 2, Appendix B), successfully eliminating $80\%$ of redundant information without compromising accuracy. Additionally, integrating a refining agent enhances reasoning depth and performance while substantially shortening the model's context window. This process is formulated as:

$$o_{refine} = F(p, q, r), \qquad (12)$$

where $p$, $q$, and $r$ denote the prompt template, the retrieved sub-question, and the tool-system's redundant output, respectively.

## 4. Experiments

### 4.1. Experimental Setup

**Implementation Details.** We employed the veRL (Sheng et al., 2025) framework for reinforcement learning, utiliz-

ing Qwen2.5-VL-7B as the backbone model and GRPO for optimization. Specific configuration details include a batch size of 128, with 8 rollouts per problem. The maximum interaction rounds were set to 14, and the model was trained for 2 epochs. For evaluating rewards based on training outcomes, we deployed Qwen3-14B (Yang et al., 2025a) locally. Our method employs low-cost local tool services during the training phase, while utilizing online search services for testing. The specific system prompts, judge prompts, and summarization prompts are presented in Appendix H.

**Datasets and Benchmarks.** For training data, we utilized the open-source FVQA-train dataset followed by our proposed deep exploration dataset BridgeVQA.

To ensure a fair and consistent comparison, our model and all baseline methods are evaluated on the same benchmarks, which collectively span knowledge-intensive Visual Question Answering (VQA) and multimodal retrieval-based reasoning challenges. These benchmarks include: FVQA-test (Wu et al., 2025), Infoseek (Chen et al., 2023), SimpleVQA (Cheng et al., 2025), MMsearch (Jiang et al., 2024), and LiveVQA (Fu et al., 2025). In the LiveVQA experiments, the latest official versions were used throughout to ensure a fair comparison. For specific datasets, can view it in the Appendix G.

To validate the generalization of our proposed SPAI method to other non-search-based visual tasks, we referred to the evaluation system in (Zhu et al., 2025b). To ensure comparability and fairness, we evaluate all models under a consistent set of evaluation protocols. we applied it to Math-Verse (Zhang et al., 2024a), MathVision (Wang et al., 2024), MathVista (Lu et al., 2023), and HallusionBench (Guan et al., 2024) and ChartQA (Masry et al., 2022).

**Baselines.** (1) The initial evaluation paradigm employed

*Table 2.* Comparison of our method with other visual tasks.

| Model | Average | MathVista | MathVerse | MathVision | ChartQA | HallBench |
|---|---|---|---|---|---|---|
| *Close-source Models* | | | | | | |
| GPT-4o | — | 63.8 | 50.8 | 30.4 | — | 55.0 |
| Claude-3.7-Sonnet | — | 66.8 | 52.0 | 41.3 | — | 55.4 |
| *Open-source Models* | | | | | | |
| Qwen2.5-VL-7B | 55.9 | 68.1 | 41.1 | 25.4 | 79.8 | 65.2 |
| OpenVLThinker-7B | 55.7 | 69.7 | 46.4 | 24.8 | 78.4 | 59.1 |
| WeThink-7B | — | 70.9 | 44.7 | 27.2 | — | 55.1 |
| ThinkLite-VL-7B | 59.6 | 72.4 | 45.2 | 28.0 | 82.0 | 70.2 |
| VL-Rethinker-7B | — | 73.7 | — | 28.4 | 79.0 | 69.9 |
| Ours (w/ SPAI) | **61.6** | **74.2** | **50.6** | **29.2** | **82.1** | **72.0** |

is direct inference, where the model operates exclusively without access to external callable tools. In this configuration, the model is constrained to rely entirely on its internal parametric knowledge for question answering. We utilized this specific setting to establish a performance benchmark for several state-of-the-art models, including GPT-4o (Hurst et al., 2024), Gemini-2.5 Pro (Team et al., 2023), and the open-source Qwen2.5-VL series. (2) The second approach involves training intelligent agents using methods such as SFT, RL, or CoT. Both methods utilize image search and text search tools. Note that our training employs low-cost local tool services. It is worth noting that the agent Web-Watcher (Geng et al., 2025) uses more paid tools (such as visit, code)

**Metrics.** For fair comparison, we employ the MMSearch-R1 methodology, utilizing the LLM-as-Judge framework as an evaluation metric to measure the accuracy of model responses. Specifically, we deploy Qwen2.5-72B-Instruct locally as the judging model, which determines correctness by receiving the original image, question, ground truth answer, and model response.

*Table 3.* Ablation experiments in different components.

| BridgeVQA | SPAI | BGAS | Refining | Acc |
|---|---|---|---|---|
| - | - | - | - | 57.7% |
| ✓ | - | - | - | 59.7% |
| ✓ | - | ✓ | - | 60.8% |
| ✓ | ✓ | - | - | 63.4% |
| ✓ | ✓ | ✓ | - | 64.9% |
| ✓ | ✓ | ✓ | ✓ | **66.8%** |

## 4.2. Main Results

**Evaluation on Search-oriented Bench.** To ensure robustness, Table 1 reports the average performance across three independent inference trials. Our method significantly outperforms both open-source and closed-source models rely-

*Table 4.* Comparison of our method and others in different visual datasets.

| Setting | FVQA-test | HallBench | Step |
|---|---|---|---|
| Ours | **66.8%** | **72.0%** | 160 |
| GRPO | 57.7% | 65.2% | 200 |
| DAPO | 64.0% | 68.7% | 225 |

*Table 5.* Ablation experiments with different specific parameter configurations. SPAI Inject represents negative sample replacement ratio of Structural Proximity-weighted Advantage Injection mechanism, BGAS represents Bidirectionally Guided Adaptive Smoothing.

| SPAI Inject | BGAS $(\mu_2, \sigma_2)$ | Acc | mean turns |
|---|---|---|---|
| 5% | (4, 1.2) | **66.8%** | 9 |
| 5% | (3, 1.2) | 65.2% | 7 |
| 5% | (0, 0) | 63.4% | 10 |
| 0% | (3, 1.2) | 63.2% | 7 |
| 10% | (4, 1.2) | 65.3% | 9 |

ing on direct reasoning. These results indicate that direct reasoning is insufficient for this task, as its knowledge-intensive nature requires the integration of external knowledge and multi-turn interactive reasoning to generate accurate solutions.

Our model achieves SOTA performance compared to other trained agents. For instance, it outperforms MMSearch-R1 by 8.4% on FVQA-test and 4.9% on InfoSeek. These gains are attributed to our mechanism's ability to enhance model exploration, enabling it to discover more complex samples to resolve challenging queries. Notably, our RL-only approach achieves a substantial improvement even over Deep-EyesV2, which utilizes a combined SFT and RL strategy; this underscores our method's capacity to reduce reliance on costly supervised data. Furthermore, in comparison to the base model Qwen2.5-VL-7B, our framework yields a re-

markable improvement of 32.6% on the FVQA-test dataset, further validating the high efficiency of our training framework. Regarding the training environment, while contemporary approaches rely on knowledge-intensive online Google Search tools, our method utilizes cost-effective local tool services and still achieves state-of-the-art (SOTA) performance.

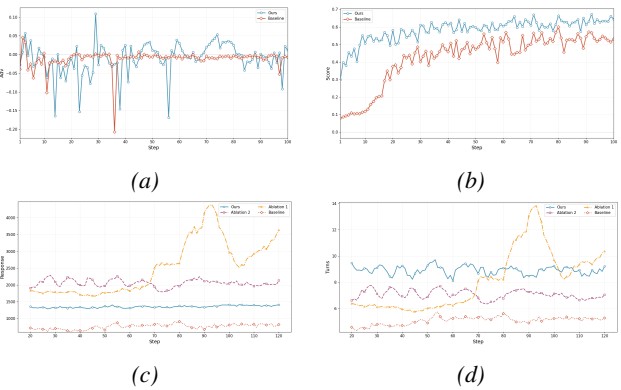

*Figure 6.* (a) Advantage distribution of GRPO and ours in FVQA. (b) Training score of GRPO and ours in FVQA. (c) Response length of GRPO and ours in FVQA. (d) Turns of GRPO and ours in FVQA.

**Evaluation Other Visual Bench.** To demonstrate the applicability of SPAI to other non-search visual tasks, we compared it against representative benchmarks in mathematics, charts, and perception. As shown in Table 2, our method achieves significant performance improvements over baseline models. For example, achieving a 9.5% improvement on the Mathverse dataset, while also demonstrating advantages over most existing methods (Yang et al., 2025b; Wang et al., 2025c;a; Deng et al., 2025; Hong et al., 2025) in current comparisons, outperforming the majority of trained approaches.

### 4.3. Ablation Study

**Component Ablation.** As shown in Table 3 and Figure 6 (b), (c), and (d), we conducted detailed ablation studies on the proposed framework. Overall, the performance of each module demonstrates steady improvement, accompanied by an effective increase in interaction turns and a significant optimization of response quality. These results successfully address the two major limitations previously identified: the framework not only mitigates exploration myopia and trajectory redundancy but also significantly enhances model performance, demonstrating the overall effectiveness of the proposed system.

Quantitatively, the multi-hop dataset improved baseline performance by 2%, underscoring its contribution to reasoning depth. Subsequent ablation of SPAI and BGAS mechanisms showed incremental gains of 3.7% and 1.5%, respec-

tively, validating the efficiency of the integrated framework. Furthermore, the combination of our dataset and mechanisms encouraged more profound exploration. Although the reward mechanism slightly distilled the response length and turn count, the results surpassed the baseline across all metrics. This step-by-step evaluation demonstrates our method's ability to balance robust exploration with redundancy control.

**Compared with Other RL algorithms.** As shown in Table 4, under the same configuration, our method achieves higher performance with fewer steps compared to other classical reinforcement learning methods. This demonstrates that the framework not only enhances performance but also accelerates exploration, effectively resolving the exploration degeneration dilemma in the current task.

**Experiment on Advantageous Signals.** As shown in Figure 5, the SPAI mechanism amplifies both positive and negative advantage values by incorporating spatial trajectory features, thereby enhancing the diversity of advantage signals. This leads to a more dispersed distribution of advantages, enabling the model to explore more effectively and avoid exploration collapse. These results demonstrate that our modeling of spatial reward trajectories successfully achieves diversified exploration. Furthermore, as illustrated in Figure 6 (a), our method ensures a sufficient supply of advantage signals throughout the training process, effectively preventing signal stagnation.

*Table 6.* Comparison of length-aware mechanisms on FVQA-test dataset.

| Method | Acc |
|---|---|
| Ours | **66.8%** |
| $L_2$ Normalization | 62.2% |
| Length Reward | 60.3% |

**Sensitivity Analysis and Hyperparameter Robustness.** As presented in Table 5, we conducted an ablation study to investigate the sensitivity of the core hyperparameters in SPAI and BGAS. First, impact of SPAI Injection Ratio $N\%$: we observe an inverted U-shape performance trend regarding the negative replacement ratio $N\%$. At $N = 0\%$, the model lacks sufficient penalty for suboptimal paths. The optimal performance is achieved at $N = 5\%$, which effectively suppresses the tail of poor trajectories. Increasing $N$ to 10% leads to a performance drop. This confirms our hypothesis that treating too many samples as hard negatives, disrupting the learning of potentially valid reasoning paths. This suggests that $N$ should be set to target statistical outliers rather than a substantial portion of the batch.

Second, regarding the parameter selection and ablation studies of BGAS. We first discuss the parameter selection, regarding the efficiency reward $\Theta_1$, we mentioned in the Intro-

duction that experiments on the baseline and MMSearch-R1 revealed that the model's tool interaction converges at 2 times. Therefore, we set $\mu_1$ to the average of 2 and used a tolerant curve with $\sigma_1$ set to 3 (where the model answers correctly without strict interaction limits). Table 5 presents the ablation study for the $\Theta_2$ parameters, which, combined with Figure 6, elucidates the empirical origin of these settings. Initially, $\mu_2$ was set to 3 because the model stabilized at 3 tool calls during the early stages, effectively outperforming the baseline. Subsequently, the final $\mu_2$ was adjusted to 4, as the refining agent shifted the optimal performance peak to exactly 4 tool calls. Furthermore, $\sigma_2$ was set to 1.2 to form a narrow curve. Since 4 calls proved optimal, this narrow distribution strictly aligns the model with this specific behavior when answering incorrectly. Regarding the tolerance threshold, we found that the model eventually stabilized at 10 interactions (5 tool calls) in the later stages (Figure 6), so we ultimately set the threshold to 6.

Regarding parameter robustness, comparing $\mu$ and $\sigma$ configurations highlights a clear trade-off between exploration and efficiency. We observe that the dynamic adjustment strategy (Row 1) substantially outperforms the static settings (Rows 2 and 4). Furthermore, parameter sensitivity analysis reveals that a low $\mu$ for incorrect samples causes under-search failures due to insufficient retrieval (Row 4). Importantly, alongside these specific findings, the remaining parameter ablations show that even our worst-performing configuration (63.2%) maintains a significant edge over the baseline (59.7%), robustly validating the efficacy of our method.

**Comparison with Length-Aware Baselines.** To validate the necessity of SPAI's structural proximity design, we compare our approach against two straightforward length-aware baselines. The first is an $L_2$ Normalization baseline (Advantage Shaping), which explicitly normalizes the advantages of trajectories sharing the same length to differentiate them based on absolute CoT length:

$$z_{i,t} = \frac{r_{i,t}}{\|\mathbf{r}_t\|_2 + \mathbb{1}_{\{\|\mathbf{r}_t\|_2 = 0\}}}. \tag{13}$$

The second is a heuristic Length Reward baseline, which explicitly provides a static reward for achieving exactly 4 tool calls. The comparative results are summarized in Table 6.

As shown in Table 6, while both the $L_2$ normalization and the static length reward provide marginal benefits, they significantly underperform SPAI. The fundamental limitation of these naive baselines is their isolated evaluation strategy: they scale rewards without considering the current trajectory's relative standing against ideal trajectories within the entire batch. Consequently, they fail to capture high-scoring, unique lengths, leading to a lack of advantage diversity.

In contrast, SPAI operates from a global perspective. By calculating structural proximity to batch-level boundaries, it not only distinguishes heterogeneous trajectories but also explicitly identifies and upweights unique, high-scoring paths. This mechanism prevents the proliferation of redundant paths and actively incentivizes the agent to explore more diverse reasoning strategies.

**Comparison with Other Visualization Methods.** As revealed by the visualization in Figure 1, our method not only effectively alleviates the repetition rate but also mitigates the rigidity of identical-reward trajectories. Furthermore, as demonstrated in Figure 4, the proposed framework successfully addresses the current issue of exploration degeneration.

## 5. Conclusion

Our proposed DR-MMSearchAgent framework effectively addresses the exploration degeneration dilemma. The framework leverages the SPAI mechanism to achieve global modeling of verification reward, thereby facilitating diversified exploration. Simultaneously, it employs the BGAS mechanism to dynamically adjust interaction tolerance, ensuring information reliability. Trained on datasets constructed from internet-based visual information, the approach achieves SOTA performance on knowledge-intensive VQA tasks.

## Acknowledgments

This work is supported by the Science and Technology Commission of Shanghai Municipality (Grant No. 25511102700), National Natural Science Foundation of China (Grant No. 62476090, 62302167, U23A20343, 62502159), Natural Science Foundation of Shanghai (Grant No. 25ZR1402135), Natural Science Foundation of Chongqing (Grant No. CSTB2023NSCQ-JQX0007, CSTB2025NSCQ-GPX0445), Open Project Program of the State Key Laboratory of CAD&CG (Grant No. A2501), Zhejiang University, Open Research Fund of Key Laboratory of Advanced Theory and Application in Statistics and Data Science-MOE, ECNU.

## Impact Statement

1. This paper presents DR-MMAgent, a framework designed to enhance the depth and reliability of multimodal search agents. 2. The datasets used in this paper, such as BridgeVQA, are sourced from publicly accessible resources, including up-to-date news outlets. Strict data processing procedures have been implemented to ensure logical consistency and full compliance with relevant privacy protection and ethical standards, with any personally sensitive information excluded.

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

# A. Theoretical Foundations

## A.1. Preliminaries

Let a trajectory be denoted by $\tau$, and let its (trajectory-level) return be:

$$R(\tau) = \sum_{t=1}^{T} r_t(\tau), \tag{14}$$

where $r_t(\tau)$ denotes the reward value of the last response token in the current trajectory. We denote the empirical mean and standard deviation of returns within a group by $\mu = mean(R)$ and $\sigma = std(R)$. The standardized GRPO advantage is defined as

$$A(\tau) = \frac{R(\tau) - \mu}{\sigma}. \tag{15}$$

And computes a SPAI score $W_i \in (0, 1)$. The injected advantage used in updates is

$$A'(\tau) = A(\tau) + A(\tau) \cdot W_i. \tag{16}$$

## A.2. The Proof of Reward-scale Invariance

In summary, this approach constructs a dimensionally decoupled decision manifold, which facilitates geometric localization within high-dimensional spaces and enables advantage estimation at the distributional level. This dual scale-invariance ensures that during the fusion of heterogeneous rewards, the physical significance of gradients remains uncontaminated by the absolute scales of numerical values.

First, show that the standardized advantage $A(\tau)$ is *exactly invariant* under affine transformations of the returns of the form $R'(\tau) = \alpha R(\tau)$ with $\alpha > 0$.

**Lemma A.1.** *Invariance of standardized advantage under affine transforms. Let $R'(\tau) = \alpha R$. Denote the batch mean and std of $R'$ by $\mu'$ and $\sigma'$. Then the standardized advantage satisfies*

$$B(\tau) = \frac{R'(\tau) - \mu'}{\sigma'} = \frac{R(\tau) - \mu}{\sigma} = A(\tau). \tag{17}$$

*Proof.* Under $R'(\tau) = \alpha R(\tau)$, we have $\mu' = \mathbb{E}[R'] = \alpha \mu, \sigma' = \sqrt{\mathbb{E}[(R' - \mu')^2]} = \alpha \sigma$. Hence:

$$B(\tau) = \frac{\alpha R(\tau) - (\alpha \mu)}{\alpha \sigma} = \frac{\alpha (R(\tau) - \mu)}{\alpha \sigma} = A(\tau). \tag{18}$$

Next, we will demonstrate the scaling invariance of the SPAI method:

*Proof.* Due to $R'(\tau) = \alpha R(\tau)$, $r'_{i,t} = \alpha r_{i,t}$,

$$z'_{i,t} = \frac{\alpha r_{i,t}}{\alpha \sqrt{\sum_{i=1}^{G} r_{i,t}^2}} = z_{i,t}. \tag{19}$$

From this, it can be seen that the normalization remains unchanged before and after, so for the subsequent steps, we obtain $W'_i = W_i$.

**Theorem A.2.** *Scale invariance of injected advantage. Under any affine transform $R'(\tau) = \alpha R(\tau)$ with $\alpha > 0$, the $A(\tau)$ and the SPAI score $W(\tau)$ are invariant, hence the injected advantage*

$$A'(\tau) = A(\tau) + A(\tau) \cdot W_i. \tag{20}$$

*is invariant as well.*

**Remark.** In practice it is common to compute a numerically-stabilized score $F(\tau) = \frac{D^-(\tau)}{D^+(\tau)+D^-(\tau)+\epsilon}$, with a small $\epsilon > 0$ to avoid division-by-zero. In practice, this occurs extremely rarely, so the impact can be considered negligible.

## A.3. Discriminability via Distributional Sparsity

Let $\tau_a$ and $\tau_b$ be two trajectories yielding identical scalar rewards $R$, ending at time steps $t_a$ and $t_b$ respectively ($t_a \neq t_b$). Under the SPAI mechanism, their injected advantages are distinct ($A'(\tau_a) \neq A'(\tau_b)$) if and only if the **reward distribution density** at these time steps differs. Specifically, if the aggregate reward norms at the two time steps are unequal:

$$\sum_{j=1}^{G} r_{j,t_a}^2 \neq \sum_{j=1}^{G} r_{j,t_b}^2, \tag{21}$$

then the trajectories are distinguishable in the latent structural space.

*Proof.* The normalized reward value for a trajectory $i$ ending at $t$ with raw reward $R$ is:

$$z_{i,t} = \frac{R}{\|\mathbf{r}_t\|_2 + \mathbb{1}_{\{\|\mathbf{r}_t\|_2=0\}}}. \tag{22}$$

Consider two trajectories $\tau_a$ and $\tau_b$ ending at $t_a$ and $t_b$ with $R_a = R_b = R$. If the distribution of valid solutions in the current batch is anisotropic, then the normalized peak values differ:

$$z_{a,t_a} \neq z_{b,t_b}. \tag{23}$$

Consequently, the virtual ideal solution components at these steps differ: $z_{t_a}^+ \neq z_{t_b}^+$. Computing the Euclidean distance

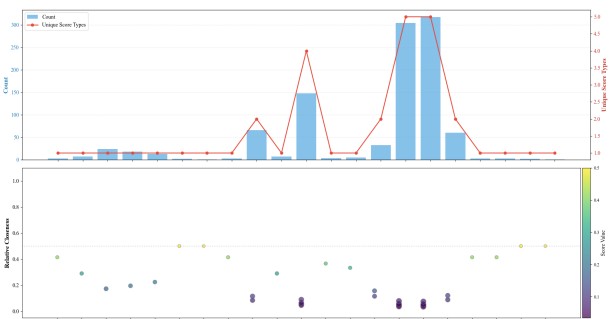

*Figure 7.* SPAI Visual Analysis. The upper figure shows the analysis of isomerism phenomenon, and the lower figure shows the scores of different rewards from a global perspective.

to the positive ideal solution $\mathbf{Z}^+$:

$$D_a^+ = \sqrt{(z_{a,t_a} - z_{t_a}^+)^2 + \sum_{t \neq t_a}(0 - z_t^+)^2} \quad (24)$$

$$D_b^+ = \sqrt{(z_{b,t_b} - z_{t_b}^+)^2 + \sum_{t \neq t_b}(0 - z_t^+)^2} \quad (25)$$

Since $z_{a,t_a}$ is the contribution to the ideal vector at $t_a$ (assuming $R$ is the max reward), the distance terms depend on the magnitudes of other components in $\mathbf{Z}^+$. Even in the simplified case where $R$ is constant, the normalization scales the entire vector dimension $t_a$ differently from $t_b$. Thus, $D_a^+ \neq D_b^+ \implies F_a \neq F_b \implies W_a \neq W_b$.

**Remark:** In practice, the distribution of reasoning steps (CoT length) typically follows a gaussian-like or long-tail distribution rather than a uniform distribution. Therefore, $\mathcal{S}_{t_a} \neq \mathcal{S}_{t_b}$ holds almost surely, enabling SPAI to adaptively weigh trajectories based on the rarity of their solution path length.

### A.4. Visualization and Validation

As shown in Figure 7, to provide experimental validation, we conducted a statistical analysis of SPAI scores at an intermediate training stage. The upper panel of the figure illustrates the phenomenon of trajectories yielding identical total rewards; it is evident that our method achieves fine-grained differentiation even when total rewards are the same. The lower panel displays the distribution of various rewards from a global perspective, demonstrating that our approach effectively discriminates advantage signals globally to facilitate diversified exploration.

### A.5. Feasibility Explanation of SPAI Design

As illustrated in Figure 1 (b), which delineates the diversity among trajectories and the emergence of completely identical paths, the positional distribution of tokens for a given reward is initially highly diverse. However, as training

progresses, this diversity steadily diminishes, leading to the suboptimal utilization of structural information during modeling. This degradation inherently motivates the adoption of SPAI, as it is specifically designed to differentiate trajectories that yield identical scalar rewards. To empirically validate this capability, we provide a statistical visualization in Appendix Figure 7. As observed in long-response scenarios, instances with identical rewards still constitute a substantial proportion of the batch (indicated by the blue bars, with over 300 trajectories sharing the same reward and advantage). Crucially, SPAI effectively breaks these ties; as shown by the red curve, our method further stratifies these previously indistinguishable trajectories into distinct advantage values, thereby restoring the granular diversity necessary for effective exploration.

## B. Context Extraction Agent

For the constructed refining agent, there is almost no training delay because it directly reuses the updatable policy model and embeds it into the entire training loop. As shown in Figure 8, this method can effectively solve the current tool redundancy problem and achieve a compression efficiency of over three-quarters.

---

**Input Query: New START treaty and number of deployed long-range nuclear warheads**

**Raw Search Results (Redundant)**

'result': [['document': 'id': '3255926', 'contents': "The 'New START' treaty is an agreement by both the US and Russian governments to limit the deploying of nuclear ballistic missiles... [Truncated 200+ words] ...An advisory opinion on this issue was originally requested by the World Health Organization (WHO) on 3 September 1993..."]]

**Word Count: ≈ 244**

↓ **Agent Refinement & Compression**

**Refined Agent Output (Concise)**

The ... effective until 2026, limits each party to no more than 1,550 strategic warheads and 700 launchers deployed, ensuring limited nuclear deployment.

**Word Count: 45 (Reduction: 82%)**

---

*Figure 8.* Comparison of information density. The agent compresses raw search results by approximately 82% while retaining all critical facts, demonstrating its effectiveness in information distillation.

## C. Computational Overhead and Deployment Feasibility.

To comprehensively evaluate the practicality of our proposed method, we analyze its computational overhead in terms of both time and memory.

During inference, the deployment costs for our method and the baselines are virtually identical. The only discrepancy arises from the tool invocation process, which introduces negligible latency. For training, we conducted a series of comparative experiments detailing the time cost multipliers, accuracy, and convergence steps, as summarized in Table 7.

*Table 7.* Comparison of computational overhead and performance.

| Method | Tool | Cost ($\times$) | Acc | Steps |
|---|---|---|---|---|
| Ours | Local | 2 | **66.8** | 160 |
| Ours | Local | 1 | 60.7 | 100 |
| GRPO | Local | 1 | 57.7 | 200 |
| GRPO | Local | 2 | 59.9 | 320 |
| MMSearch-R1 | Google | 12 | 58.4 | — |

As demonstrated in Table 7, although our method requires slightly more time per step, it consistently outperforms standard GRPO under identical total training time budgets (e.g., comparing configurations with a $1\times$ or $2\times$ time cost multiplier). Furthermore, by utilizing a free local knowledge base, our method achieves a $6\times$ efficiency improvement over MMSearch-R1—which relies on expensive online searches (Google)—while yielding significantly superior accuracy ($66.8\%$ vs. $58.4\%$). This fully demonstrates the efficiency and effectiveness of our design.

Regarding memory consumption, our training infrastructure consists of eight NVIDIA H200 GPUs (1600 GB total VRAM). Overall GPU memory utilization remains stable at approximately $90\%$, with our method introducing a negligible $0.5\%$ memory overhead compared to standard GRPO. In summary, our approach delivers high performance with minimal additional costs, making it highly feasible for real-world deployment.

## D. BridgeVQA Information Statistics

As shown in Figure 9, we visualized the constructed complex dataset and found that its sources are quite diverse. The most frequent category related to sports has only 138 entries, with 875 entries having a frequency greater than 5, and the remaining 2727 entries.

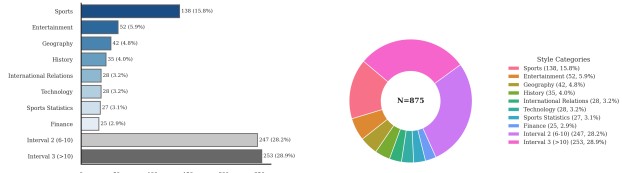

*Figure 9.* Statistical information of BridgeVQA. It shows the diversity sources of the dataset, with 875 sources appearing more than 5 times, and the remaining 2727 sources.

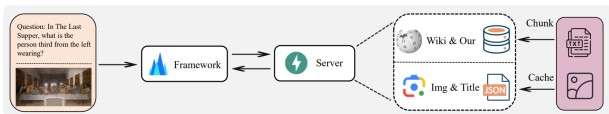

*Figure 10.* The proposed updatable tool framework. Our tool services include text search and image search. The text search knowledge base is composed of sliced web page information and wikis, while the image search is powered by Google Search and stored in a cached format.

## E. Difficulty test of BridgeVQA

We conducted a test on the difficulty level of BridgeVQA, it does not overlap with the training set.. Specifically, we allocated another part as the test set. As shown in Table 8, compared to existing datasets, it can be found that BridgeVQA is relatively difficult, and our dataset can definitely guarantee correct answers because it comes from real web pages.

*Table 8.* Performance Comparison on FVQA-test and BridgeVQA Datasets

| Model | FVQA-test | BridgeVQA |
|---|---|---|
| *Direct* | | |
| Qwen2.5-VL-7B | 20.9 | 9.6 |
| Qwen2.5-VL-32B | 24.7 | 13.4 |
| *Search Agent* | | |
| Qwen2.5-VL-7B | 34.2 | 23.9 |
| Qwen2.5-VL-32B | 51.3 | 34.2 |
| MMSearch-R1 | 58.4 | 35.3 |
| **Ours** | **66.8** | **44.6** |

*Table 9.* Performance and cost-efficiency trade-off, measured after one epoch of training.

| Tools | Acc | Cost | Call Time (s) |
|---|---|---|---|
| Google | 52.7% | $0.50 / 1k calls | 6017 |
| Ours | **50.0%** | **Virtually $0.00** | **457** |

## F. The Verification of Training Tool Service

To verify the advantages of local upgradable tool services, we validated their scalability and low-cost reliable training

*Table 10.* Comparison under Identical Evaluation Criteria.

| EM | LLM-as-Judge | MMSearch-R1 | Ours |
|:---:|:---:|:---:|:---:|
| ✓ | | 42.5% | 48.2% |
| | ✓ | 44.3% | 50.0% |

features. Specifically, from Figure 10, it can be observed that our tool is decoupled from the training environment. By slicing the textual information of the constructed data and pre retrieving and caching web page images, a tool service is ultimately formed. This process can be used to construct any text and image information, achieving a closed loop from data to tools. From Table 9, it can be seen that our training tool service has good training efficiency and is faster in search speed than online services. Furthermore, we have achieved very low cost consumption and low precision loss.

## G. Specific Information

To ensure a fair and rigorous head-to-head comparison, our evaluation strictly adheres to the standardized data partitions used in previous state-of-the-art works. Specifically, for Infoseek, we evaluate on the same curated subset of 2,000 samples. For SimpleVQA, we restrict our analysis to the 1,013 English-only QA pairs to maintain consistency. Our evaluation on MMSearch is conducted on the selection of 171 image-based samples, and for LiveVQA, we utilize the 2,384-sample version currently available to align with the latest reported benchmarks.

As illustrated in Table 10, we present a fair comparison under unified evaluation protocols. It is evident that our approach consistently achieves superior performance across both Exact Match (EM) and LLM-as-Judge metrics.

## H. Prompt

We have listed system prompt, summarized prompt and judge prompt.

**Step 1: Think**
- **This is the starting point for every turn.**
- Carefully analyze the user's query, break down the problem, and formulate a plan.
- Your entire reasoning process must be enclosed in `<think>...</think>` tags.

**Step 2: Act (Tool Call)**
- If your plan requires information you don't have, call **one single tool**.
- The tool call must be enclosed in `<tool_call>...</tool_call>` tags.
- If you can answer without tools, **skip this step** and move directly to Step 4.

**Step 3: Observe and Think Again**
- This is a **critical step**. Receive the tool's observation and analyze it.
- You **MUST** start a new thought process in `<think>...</think>` tags to analyze this output.
- **Decide:** If information is still insufficient, return to **Step 2**. If sufficient, proceed to **Step 4**.

**Step 4: Summarize**
- **Execute only when the decision in Step 3 is to provide the final answer.**
- Summarize gathered info in `<summary>...</summary>` tags:
  1. **Extract key points**: Identify and condense relevant information.
  2. **Eliminate redundancy**: Remove repetitive details for clarity.
  3. **Verify consistency**: Ensure alignment with query and avoid hallucinations.

**Step 5: Answer**
- Formulate the final answer based on the Step 4 summary.
- **Constraint:** The answer must **not exceed 30 words**.
- Enclose your final answer in `<answer>...</answer>` tags.

### System Prompt for Agent

You are a helpful assistant. You can call functions to assist with the user query.
**Important:** You must call only one function at a time. After each function call, wait for the execution result before making the next function call if needed.
**# Tools**
You are provided with function signatures within `<tools></tools>` XML tags:
**Workflow and Output Format**
You must follow these steps in order. Every conversation turn should start from Step 1.

### Summary Prompt for Extract Agent

**\*\*Role\*\***: You are a professional and concise information synthesizer.
**\*\*Task\*\***: Synthesize a \*\*key findings summary\*\* relevant to the "Question" based on the "Evidence/Reasoning" (i.e., the collection of search results).
**\*\*Constraints\*\***: The summary must be \*\*strictly limited to under 50 words\*\*. Do not output any titles, roles, extra separators, or explanatory text. Begin the summary directly.
**\*\*Input\*\***: - Question: $searchquery$ - Evidence/Reasoning: $message$

**Output Format**: Output only a single, concise summary text, not exceeding 50 words.

## Judge Prompt for Answer Evaluation

Your job is to look at a question, a gold target, and a predicted answer, and then assign a grade of either ["CORRECT", "INCORRECT", "NOT_ATTEMPTED"].

**1. CORRECT Predicted Answers**
- **Criteria:** Answers must fully contain the gold target information without contradicting it. Semantic meaning matters; capitalization, punctuation, and grammar do not. Hedging is allowed if the target is included.
- **Examples:**

    Q: Obama's children? — Gold: Malia and Sasha
    A1: sasha and malia obama
    A2: Malia Ann and Natasha Marian, commonly referred to as Malia and Sasha...

**2. INCORRECT Predicted Answers**
- **Criteria:** Any factual statement contradicts the gold target. Hedged incorrect statements (e.g., "it is possible that...") are still incorrect.
- **Examples:**

    Gold: Malia and Sasha
    A1: Malia, Sasha, and Susan. (Extra info contradicts facts)
    A2: Obama has three children. (Factual error)

**3. NOT_ATTEMPTED Predicted Answers**
- **Criteria:** The important information is missing, but no statements contradict the gold target.
- **Examples:**

    A1: I don't know.
    A2: I know one is Malia, but I'm not sure about the other.

**Grading Task**
Simply reply with the letter corresponding to the grade. Do not apologize or add extra text.
```
Question: {question}
Gold target: {correct_answer}
Predicted answer: {response}
```
**Options:**
- **A:** CORRECT
- **B:** INCORRECT
- **C:** NOT_ATTEMPTED

**Output:** Return only "A", "B", or "C".

# I. Case Study

## An Example of BridgeVQA

❷ **Question:** What is the name of the horseback riding event that the woman who celebrated her 83rd birthday with the man in the image attended, while both wore matching helmets, black jackets, and white pants?

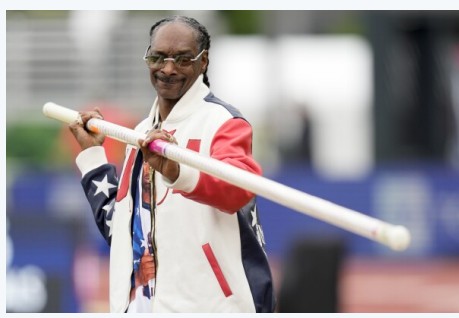

**CoT:**
1. [Visual Step] Identify the image shows 'the man in the image'. 2. [Text Search 1] Search for 'the man in the image' + '83rd birthday'. Found linkage to Martha Stewart celebrating her 83rd birthday with Snoop Dogg in paragraph [X]. 3. [Text Search 2] Now search for 'Martha Stewart horseback riding event' to find the specific detail requested. Found 'attended a equestrian team dressage event' in paragraph [Y]. 4. [Conclusion] The answer is Dressage Team Grand Prix Final.

**<answer> Dressage Team Grand Prix Final. </answer>**

## A Case of FVQA

❷ **Question:** Where was the latest festival for impressive traditional Hina dolls hosted?

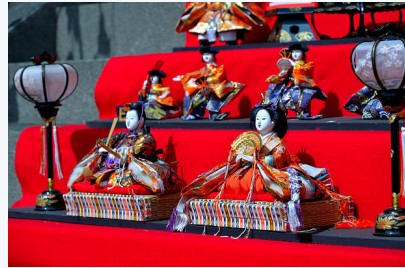

**Step 1**
<think> The images and titles provided are related to vintage Hina dolls, but they do not directly indicate the location of the latest festival for these dolls. We need to

focus on finding information about recent festivals featuring traditional Hina dolls. `</think>`

**Tool Call:** `web_image_to_image_search`

> 👁 Tool Response
>
> **Results:** 1. Vintage Japanese Hina Doll... 2. Traditional Folk Craft... 3. Girls' Festival dolls...
> **Retrieved Images:**
>
> 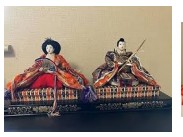 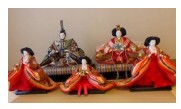 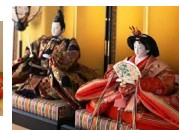

**Step 2**

`<think>` The information provided by the Wikipedia search indicates that the latest festival for impressive traditional Hina dolls was held in Tokyo, at specific venues like Inatori Onsen and Hotel Gajoen Tokyo. `</think>`

**Tool Call:** `web_text_search(query="Where was the latest festival for impressive traditional Hina dolls hosted?")`

> ☰ Tool Response
>
> **Observation:** The latest festival for impressive traditional Hina dolls was hosted in **Tokyo**, specifically at **Inatori Onsen** and at **Hotel Gajoen Tokyo**.

**Step 3**

`<think>` The response confirms that the most recent Hina Matsuri was held on March 3, 2025. It is traditionally celebrated annually on this date in Japan. `</think>`

**Tool Call:** `web_text_search(query="When did the most recent Hina Matsuri take place?")`

> ☰ Tool Response
>
> **Observation:** The most recent Hina Matsuri (Girl's Day) took place on **March 3, 2025**.

---

`<summary>` The latest festival for impressive traditional Hina dolls... The most recent Hina Matsuri took place on March 3, 2025, in Japan. `</summary>`

✅ **`<answer>` The latest festival for impressive traditional Hina dolls was hosted in Tokyo, specifically at Inatori Onsen and Hotel Gajoen Tokyo, during the most recent Hina Matsuri on March 3, 2025. This event is part of the annual celebration of Girl's Day in Japan. `</answer>`**

