# OpenReview forum: "DR-MMSearchAgent: Deepening Reasoning in Multimodal Search Agents"
_ICML.cc/2026/Conference — ICML 2026 regular_

### Official Review · Reviewer_4P4d · 2026-03-12

**Soundness:** 3
**Presentation:** 2
**Significance:** 3
**Originality:** 3
**Overall Recommendation:** 4
**Confidence:** 3

**Summary:**

In this paper, the authors try to enhance the multi-turn interactive reasoning capabilities of MLLMs, especially in search domain. The authors propose DR-MMSearchAgent framework which is composed of (1) the SPAI mechanism that constructs virtual positive and negative ideal solutions based on the token-level position of the terminal reward; and (2) BGAS technique shifts between an efficiency-first regime and an exploration-driven regime. Extensive experiments demonstrate that the proposed method achieves SOTA performance on knowledge-intensive benchmarks like FVQA-test and InfoSeek, outperforming strong baselines such as MMSearch-R1.

**Compliance With Llm Reviewing Policy:**

Affirmed.

**Key Questions For Authors:**

Please refer to the paper's weakness. I think more clarification of the designed method would help the readers understanding the article.

**Limitations:**

yes

**Strengths And Weaknesses:**

**Strengths:** The performance gains are substantial. Furthermore, the ablation studies are thorough, and the application of SPAI to non-search visual tasks (MathVerse, ChartQA) proves the generalizability of the proposed advantage injection method.

**Weakness:**
1. In the SPAI formulation, the reward matrix $R'$ is constructed such that only the last token position contains a non-zero value, while all other positions are zero. For long-context reasoning, the maximum sequence length can be extremely large. Calculating Euclidean distances in such a highly sparse and high-dimensional space is mathematically risky due to the curse of dimensionality. Could the author further clarify this design?
2. The proposed DR-MMSearchAgent framework seems to rely heavily on heuristic and empirically derived hyperparameters (SPAI Threshold, BGAS Parameters, Reward Weighting).  While these specific values clearly yield optimal performance on the evaluated benchmarks (e.g., BridgeVQA, FVQA), they may overfit the statistical priors of these specific datasets or the specific model scale used. Could the author clarify the selection criteria of these hyperparameters?
3. Evaluation on harder benchmarks. The authors claim that the proposed methods can handle more difficult tasks (i.e., may require longer trajectories). I wonder about the evaluation results on harder benchmarks such as MM-BrowseComp.

---

> ### Author Rebuttal · Authors · 2026-03-29
>
> We greatly appreciate your positive evaluation and hope the following responses will continue to earn your support.
>
> > ` Weaknesses 1: SPAI design`
> * Thank you for your question. We conducted detailed experimental statistics, as shown in **Figure 1(b)**, which separately displays the diversity of identical trajectories and completely identical trajectory situations. We found that initially, the positional information for the same reward is highly diverse (left), but as training progresses, this diversity begins to decline, leading to underutilization when modeling this information.
> * Therefore, we chose SPAI, as our method can further differentiate between identical rewards. To verify our hypothesis, we provided a statistical visualization of the method in **Appendix Figure 7** (page 12). As can be observed, in long-response scenarios, cases with identical rewards (refer to the blue bars) still account for a relatively large proportion (with over 300 trajectories sharing the same reward and advantage). Our method can further differentiate these identical trajectories (as indicated by the red curve, which further distinguishes five distinct advantage types).
> * This is also further validated by **Figure 5** (page 6). Overall, we demonstrate that our method is entirely feasible.
>
> > ` Weaknesses 2: Hyperparameter selection criteria`
> * For the motivation behind selecting the hyperparameters $\mu$ and $\sigma$, please refer to our response to **Reviewer Y3Xk's Weaknesses 2 & Key Questions 2**, which provides a detailed and rigorous selection process.
> * The basis for the threshold selection is found in **Appendix Figure 7** (page 12). Statistical analysis shows that the lowest reward function accounts for about 5%, and the highest reward also accounts for 5%. We also conducted an ablation study in **Table 5** (page 7), which fully verifies that this threshold can maximize the suppression of incorrect paths.
> * Regarding the reward function weight settings, we drew inspiration from current multimodal agent methods in the field, assigning 0.8 for accuracy and 0.2 for format. Since we use pure RL without SFT, we must ensure the format is correct. Therefore, we further divided the outcome reward weights, resulting in the current allocation.
>
> > ` Weaknesses3: Evaluation on MM-BrowseComp`
> * We present the experiments based on your suggestion (using the 243 questions from the latest version of the dataset that only contain images):
> | Method | Overall Accuracy |
> | :--- | :--- |
> | Ours | **5.96** |
> | MMsearch-R1 | 0.85 |
> | Qwen2.5-VL-7B-Instruct | 0 |
> | Qwen2.5-VL-72B-Instruct | 0.45 |
> | DeerFlow(Qwen2.5-VL-72B-Instruct) | 1.85 |
> | DeerFlow(GPT-4o) | 1.85 |
>
> * The comparison in the table shows that our method holds a significant advantage, and it has outperformed almost all open-source agents in MM-BrowseComp.
> * It is worth noting that we use low-cost local search for training; with an adequate budget, using online search for training would yield even better results. This further confirms that our method can conduct deep exploration, combining low cost with high performance.
>
> > ` Key Questions: `
> * Sincerely thank you for this suggestion. In the revised version, we will consolidate the currently dispersed ablation studies and appendix materials into a more unified methodological description to ensure a clearer and more cohesive presentation for the readers.

---

> > ### Author Rebuttal · Reviewer_4P4d · 2026-04-03
> >
> > The authors address my concerns during the rebuttal phase. I'll keep my relative positive score.

---

> > > ### Author Response · Authors · 2026-04-03
> > >
> > > Thank you very much for your recognition of our rebuttal. We are glad that our response has addressed your concerns.

---

### Official Review · Reviewer_JvXT · 2026-03-12

**Soundness:** 3
**Presentation:** 2
**Significance:** 3
**Originality:** 3
**Overall Recommendation:** 4
**Confidence:** 4

**Summary:**

This paper proposes DR-MMSearchAgent (Deepening Reasoning MMSearchAgent), a framework to improve reasoning capabilities and reliability of multimodal search agents. The paper identifies two key problems with existing RL-based multimodal search methods like MMSearch-R1 (Wu et al., 2025): (1) terminal rewards appended to the last token prevent the advantage function from distinguishing trajectories with exploratory behavior, leading to premature interaction collapse with only two-turn tool interactions; and (2) excessively redundant context hinders agents from absorbing useful feedback. To address these issues, the authors propose two main technical contributions: (a) structural proximity-based advantage estimation that derives advantage signals from the whole rollout trajectories in a batch, encouraging diverse trajectory lengths even when containing the same correct answer; and (b) differentiated Gaussian rewards that dynamically calibrate interaction tolerance to ensure information reliability and reduce redundancy. The authors also construct a multi-step deep-reasoning dataset containing 3,602 high-quality QA pairs with at least 3 reasoning steps. Experiments demonstrate state-of-the-art performance, with improvements of 8.4 points over MMSearch-R1 on FVQA-test, along with consistent gains across benchmarks including MMSearch, InfoSeek, LiveVQA, and SimpleVQA.

**Compliance With Llm Reviewing Policy:**

Affirmed.

**Final Justification:**

The authors rebuttal has fully resolved my concerns, so I decide to raise my score to 4.

**Key Questions For Authors:**

1. How sensitive is the method to the hyperparameters of the differentiated Gaussian rewards (μ, σ, tolerance thresholds)? An ablation varying these parameters would help assess the robustness of the approach. A positive result here would increase my confidence in the method's practical applicability.

2. Could you provide comparisons against simpler reward shaping baselines (e.g., length-based bonuses, RUDDER-style reward redistribution)? Demonstrating clear advantages over such alternatives would significantly strengthen the contribution and could raise my overall assessment.

3. Could you provide a more detailed analysis of how the average number of search interactions evolves during training? Understanding whether the model stabilizes to a particular interaction depth and whether this varies across query types would help validate the claim that the method addresses premature interaction collapse effectively.

4. What is the computational overhead of the structural proximity computation relative to standard GRPO training? Understanding the wall-clock time and memory costs would help assess practical deployment feasibility.

**Limitations:**

Yes.

**Strengths And Weaknesses:**

Strengths:

1. The paper identifies a meaningful and well-motivated problem: premature interaction collapse in RL-trained multimodal search agents. This observation is empirically grounded and provides a useful insight for the community working on agentic multimodal systems.

2. The differentiated Gaussian reward mechanism is a reasonable design choice for dynamically calibrating interaction tolerance. The idea of penalizing redundant interactions while encouraging informative ones addresses a practical challenge in multi-turn search agents.

3. The experimental results are comprehensive, covering multiple benchmarks (MMSearch, FVQA, InfoSeek, LiveVQA, SimpleVQA) and showing consistent improvements over the MMSearch-R1 baseline. The 8.4-point improvement on FVQA-test is notable. The inclusion of ablation studies examining individual components is appreciated.

Weaknesses:

1. The paper would benefit from comparisons against simpler baseline alternatives for encouraging deeper search interactions, such as length-based reward bonuses or standard reward redistribution techniques from the RL literature (e.g., RUDDER). Without these comparisons, it is difficult to fully assess whether the structural proximity mechanism provides benefits beyond simpler approaches.

2. The presentation could be improved in several areas. Key concepts like "structural proximity" would benefit from earlier and more formal definition. An algorithmic pseudocode description would aid reproducibility. The discussion of concurrent related works (e.g., M³Searcher, REDSearcher) could be more thorough.

3. The computational overhead of the structural proximity computation, which involves comparing trajectories within a batch, is not discussed. Understanding the scalability characteristics of the method would be important for assessing its practical applicability.

---

> ### Author Rebuttal · Authors · 2026-03-29
>
> We sincerely thank you for your positive evaluation of our motivation, methodology, and experiments. We believe the following responses will address all your concerns.
>
> > ` Weaknesses 1 & Key Questions 2: Comparison with simpler reward shaping baselines`
> * Regarding simpler baseline comparisons, we have added two comparative methods: fixed length reward and advantage shaping.
> | Method | Acc (FVQA-test) |
> | :--- | :--- |
> | Ours | **66.8** |
> | 4 Calls (Length Reward) | 60.3 |
> | L2 Normalization (Advantage Shaping) | 62.2 |
> | BGAS only (Length Reward) | 60.8 |
> | SAPI (Advantage Shaping) | 63.4 |
>
> * As shown in the table, our module achieves a significant lead over any comparative methods (the last two rows are from **Table 3**, page 7). This indicates that mechanically increasing model calls does not help the model deepen the reason capabilities.
>
> > ` Weaknesses 3 & Key Questions 4: Computational overhead and practical deployment feasibility`
>
> * Regarding time costs: during inference, the deployment costs for both are nearly identical; the only difference is the tool invocation time, which is negligible. For training, we conducted the following comparative experiments:
> | Method | Training Tool | Time Cost (Multiplier) | Acc | Step |
> | :--- | :--- | :--- | :--- | :--- |
> | Ours | local | 2 | **66.8** | 160 |
> | Ours | local | 1 | 60.7 | 100 |
> | GRPO | local | 1 | 57.7 | 200 |
> | GRPO | local | 2 | 59.9 | 320 |
> | MMsearch-R1 | google | 12 | 58.4 | — |
>
> * As shown in the table, although our method takes more time per step, under the same total training time cost (the second row and the fifth row), our method consistently outperforms standard GRPO. Furthermore, when training with a free local knowledge base, our efficiency is 6 times higher than MMsearch-R1's expensive online search, with significantly improved results, fully demonstrating the effectiveness of our method.
> * Regarding memory, we use 8 H200 GPUs with 1600GB. The GPU memory utilization remains around 90%, and our method introduces a negligible memory overhead of only 0.5% compared to GRPO. Overall, our method features low cost and high performance, making it highly feasible for actual deployment.
>
> > ` Key Questions 1: Sensitivity to hyperparameters and robustness ablation`
>
> * For the selection of the hyperparameters $\mu$ and $\sigma$, please refer to our response to **Reviewer Y3Xk's Weaknesses 2 & Key Questions 2**.
> * **Table 3** and **5** present the ablation studies of hyperparameters:
> | SPAI Inject | BGAS ($\mu_2, \sigma_2$) | Acc |
> | :--- | :--- | :--- |
> | 5% | (4, 1.2) | 66.8% |
> | 5% | (3, 1.2) | 65.2% |
> | 5% | (0, 0) | 63.4% |
> | 0% | (3, 1.2) | 63.2% |
> | 10% | (4, 1.2) | 65.3% |
> | GRPO (0%) | (0, 0) | 59.7%  **(lowest)** |
>
> * Our method consistently outperforms the baselines. Furthermore, regarding the tolerance threshold, we found that the model eventually stabilized at 10 interactions (5 tool calls) in the later stages (Figure 6), so we ultimately set the threshold to 6.
>
> > ` Key Questions 3: Evolution of search interactions and prevention of premature collapse`
> * **Figure 6(d)** (page 7) presents a detailed ablation study on the total number of interaction rounds. Our method engages in 8-10 interactions, continues to explore in the later stages, and steadily improves in performance. In contrast, the baseline stabilizes at 5 interactions after 90 steps, with only slow performance gains. Our method can distinguish heterogeneous trajectories sharing the same reward and upweight the highest-scoring unique paths. This reduces redundancy and drives deeper model exploration. We use a free local knowledge base during training and online search during inference. Performance remains stable across different numbers of interactions and is significantly higher than that of methods ( MMsearch-R1) trained entirely online.
>
> > ` Weaknesses 2: Pseudocode, and related works`
> * Thank you for your suggestions. We will include the pseudocode for SAPI and release all data and code in the subsequent version to aid reproducibility.
> * Regarding the related work you mentioned: the former primarily focuses on MRAG, whereas we utilize a free local image search, which offers lower usage costs and higher performance. The latter is mainly focused on text reasoning, using SFT and RL to enhance single-modality capabilities, whereas we achieve state-of-the-art performance using only RL.

---

> > ### Author Rebuttal · Reviewer_JvXT · 2026-04-04
> >
> > The authors address all of my concerns. I am considering raising my score.

---

> > > ### Author Response · Authors · 2026-04-04
> > >
> > > We sincerely appreciate your support for our work and are glad that our rebuttal has addressed all your concerns. Thank you again for the time and effort you have dedicated to reviewing this paper.

---

### Official Review · Reviewer_Y3Xk · 2026-03-14

**Soundness:** 2
**Presentation:** 2
**Significance:** 3
**Originality:** 2
**Overall Recommendation:** 4
**Confidence:** 3

**Summary:**

This paper proposes DR-MMSearchAgent to address exploration degradation in RL-trained multimodal search agents, via a TOPSIS-inspired advantage reweighting mechanism (SPAI), a Gaussian adaptive reward (BGAS), and a new multi-hop VQA dataset (BridgeVQA). Strong results are shown on knowledge-intensive VQA benchmarks.

**Compliance With Llm Reviewing Policy:**

Affirmed.

**Final Justification:**

The rebuttal addressed my main concern. The added baseline comparison (SPAI 66.8 vs Normalization 62.2 vs Length Reward 60.3) addresses my main concern. The other clarifications are also appreciated. I raise my score from 3 to 4.

**Key Questions For Authors:**

1. How does SPAI compare to a simpler length-dependent advantage bonus?
2. Could you clarify how the single $(\mu, \sigma)$ entries in Table 5 map to the two regimes $\Theta_1$ and $\Theta_2$?
3. What explains the gap with DeepEyes2 on MMSearch (61.0 vs 63.7)?

**Limitations:**

Data ethics are discussed in the Impact Statement. A brief note on method-level limitations (e.g., scalability beyond the 7B backbone, dependence on the refining agent) would strengthen the paper.

**Strengths And Weaknesses:**

**Strengths:**
1. Well-motivated problem with clear empirical evidence of agent exploration collapse.
2. Consistent gains across five benchmarks and clean ablation studies.

**Weaknesses:**
1. SPAI introduces notable complexity (normalization, ideal solutions, Euclidean distances from TOPSIS), but no comparison to a simpler length-aware advantage baseline is provided to justify this design choice.
2. Presentation can be improved: some notation is dense, and minor typos remain (e.g., "reasonning" in the abstract). For BGAS, the text describes two regimes $\Theta_1 = (\mu_1, \sigma_1)$ and $\Theta_2 = (\mu_2, \sigma_2)$, but Table 5 only reports a single $(\mu, \sigma)$ per row, making the dual-regime setup hard to follow.

---

> ### Author Rebuttal · Authors · 2026-03-29
>
> We sincerely thank you for your positive evaluation of our motivation and experiments, as well as your rigorous feedback and suggestions. We hope our responses address your concerns.
> > ` Weaknesses 1 & Key Questions 1: Comparison with simpler length-aware baselines`
> * For comparison, we implemented a simpler $L_2$ Length-Aware Advantage baseline. This method normalizes trajectories of the same length to differentiate their advantages (scaling the advantage directly based on the absolute CoT length of each trajectory):
> $$
> z_{i,t} = \frac{r_{i,t}}{\sqrt{\sum_{j=1}^G r_{j,t}^2} + 1_{ \{ \sqrt{\sum_{j=1}^G r_{j,t}^2} = 0 \} }}
> $$
> Additionally, we included a simpler length reward baseline to further validate our design:
> | Method | Acc (FVQA-test) |
> | :--- | :--- |
> | Ours | **66.8** |
> | $L_2$ Normalization (Advantage Shaping) | 62.2 |
> | 4 Tool Calls (Length Reward) | 60.3 |
> * The $L_2$ baseline and length reward both provides slight benefit. But it fails to consider the current trajectory relative to ideal trajectories within the entire batch, completely ignoring high scores, unique lengths, and the resulting advantage diversity.
> * SPAI’s design is necessary because it evaluates trajectories from a global perspective. By calculating structural proximity to batch-level boundaries, it not only distinguishes heterogeneous trajectories, but also identifies and upweights the highest-scoring, unique trajectories. This actively prevents the model from outputting identical, redundant paths, forcing it to dig deeper and explore further.
>
> > ` Weaknesses 2 & Key Questions 2: Presentation & Hyperparameter mapping in table 5`
> * Regarding the efficiency reward $\Theta_1$, we mentioned in the **Introduction** (page 1) that experiments on the baseline and mmsearch-r1 revealed that the model's tool interaction converges at 2 times. Therefore, we set $\mu_1$ to the average of 2 and used a tolerant curve with $\sigma_1$ set to 3 (where the model answers correctly without strict interaction limits).
>
> * **Table 5** presents the ablation study for the $\Theta_2$ parameters. Combined with **Figure 6**, it explains the empirical origin of these settings:
> | Parameter Setting | Empirical Observation & Rationale |
> | :--- | :--- |
> | **Initial $\mu_2 = 3$** | The model stabilized at 3 tool calls in the early stage, outperforming the baseline. |
> | **Final $\mu_2 = 4$** | The refining agent shifted the optimal performance peak to 4 tool calls (**Fig. 6d**). |
> | **$\sigma_2 = 1.2$** (narrow curve) | 4 calls proved optimal. The narrow curve strictly aligns the model with this behavior when answering incorrectly. |
>
> * Furthermore, the other parameter ablations in Table 5 demonstrate that even our worst-performing configuration (63.2%) significantly outperforms the baseline (59.7%), confirming the robustness of our method (more details please refer to our response to **Reviewer JvXT 's Key Questions 1**).
> .
> * The parameter settings for $\Theta_1$ and $\Theta_2$ are illustrated in **Figure 2** . We will provide a clearer description of these settings in the next version.
>
> > ` Key Questions 3: Performance gap with DeepEyes2 on MMSearch`
> * We use a free local knowledge base for training, whereas DeepEyes2 relies on high-cost commercial online search APIs. Furthermore, **Appendix Table 7** presents the results of a preliminary online training phase. The clear upward trend indicates a strong potential to surpass the baseline if training were extended.
> * Regarding data volume, our work establishes a substantial lead over existing large-scale benchmarks (>1,000 queries). In contrast, MMSearch is a small-scale dataset with only **171** questions. Overall, we believe that a slight performance gap on a single lightweight dataset is a reasonable trade-off.
>
> > ` Limitations: Scalability and dependence on the refining agent`
> * Thank you for your suggestion. We compared our method with the baseline on Qwen3-VL-8B-Instruct. Under the same configuration, the results for the first 70 steps were **51%** and **46%**, respectively, verifying the scalability of our model.
> * Regarding the refining agent, we used a local knowledge base where the returned top-3 results contain irrelevant information; the refining agent was introduced to prevent this noise from interfering with the model.
> * Meanwhile, Appendix table 7 (page 13) presents a comparison with and without the refining agent:
> | Training Tool | Refining | Acc |
> | :--- | :--- | :--- |
> | Google | no | 52.7 |
> | Local | yes | 50.0 |
>
> * The expensive online search method without a refining agent still outperforms the local search utilizing a refining agent by 2.7%. This demonstrates that the reliance on a refining agent can be completely eliminated in real-world scenarios.

---

> > ### Author Rebuttal · Reviewer_Y3Xk · 2026-04-05
> >
> > Thank you for the detailed response. The added baseline comparison (SPAI 66.8 vs Normalization 62.2 vs Length Reward 60.3) addresses my main concern. The other clarifications are also appreciated. I am willing to raise my score to 4.

---

> > > ### Author Response · Authors · 2026-04-06
> > >
> > > Thank you for the positive feedback. We are glad our response resolved your main concerns.

---

### Decision · Program_Chairs · 2026-04-30

**Decision:**

Accept (regular)

**Comment:**

In this paper, the authors proposed DR-MMSearchAgent framework which is composed of the SPAI mechanism that constructs virtual positive and negative ideal solutions based on the token-level position of the terminal reward; and BGAS technique shifts between an efficiency-first regime and an exploration-driven regime. Originally, reviewers have concerns such as the experimental comparison, presentation of the paper, etc. After the rebuttal, all reviewers said that their concerns were well resolved by the authors, and recommended acceptance of the paper. The AC carefully read the paper, the rebuttal, and the reviewer discussions, and think the paper has good contrubtion to the community; and thus recommends acceptance of the paper. The AC also suggest the authors to further refine the presentation of the paper.